# 4SC-202 as a Potential Treatment for the Pediatric Brain Tumor Medulloblastoma

**DOI:** 10.3390/brainsci7110147

**Published:** 2017-11-03

**Authors:** Shanta M. Messerli, Mariah M. Hoffman, Etienne Z. Gnimpieba, Hella Kohlhof, Ratan D. Bhardwaj

**Affiliations:** 1Sanford Children’s Health Research Center, Department of Pediatrics, Cancer Biology and Immunotherapies, Sanford Research, Sanford School of Medicine, University of South Dakota, Vermillion, SD 57069, USA; ratan.bhardwaj@gmail.com; 2Biomedical Engineering Department, University of South Dakota, Vermillion, SD 57069, USA; Mariah.Hoffman@coyotes.usd.edu (M.M.H.); Etienne.Gnimpieba@usd.edu (E.Z.G.); 34SC AG, Fraunhoferstraße 22, 82152 Planegg, Germany; hella.kohlhof@immunic.de; 4Immunic AG, Am Klopferspitz 19, 82152 Planegg, Germany

**Keywords:** 4SC-202, medulloblastoma, pediatric brain tumor, microarray, bioinformatics

## Abstract

This project involves an examination of the effect of the small molecule inhibitor 4SC-202 on the growth of the pediatric brain cancer medulloblastoma. The small molecule inhibitor 4SC-202 significantly inhibits the viability of the pediatric desmoplastic cerebellar human medulloblastoma cell line DAOY, with an IC_50_ = 58.1 nM, but does not affect the viability of noncancerous neural stem cells (NSC). 4SC-202 exposure inhibits hedgehog expression in the DAOY cell line. Furthermore, microarray analysis of human medulloblastoma patient tumors indicate significant upregulation of key targets in the Hedgehog signaling pathway and Protein Tyrosine Kinase (PTK7).

## 1. Introduction

Medulloblastomas (MBs) and astrocytomas are some of the brain tumors most frequently found in children [1]. However, there are few effectual treatments and surgery on children is not effective as the removal of the entire brain tumor is difficult. Recently, transcriptome profiling of malignancies of the central nervous system (CNS) have provided important insights into potential targets for therapies [2,3,4]. Here, we examine how transcriptome profiling of pediatric medulloblastomas reveals targets for potential therapies.

In addition, we demonstrate how a novel orally available benzamide-type HDAC inhibitor, termed 4SC-202, significantly reduces the viability of medulloblastoma in culture, and lay the groundwork for potential preclinical work by using 4SC-202 in preclinical models of pediatric medulloblastoma. 4SC-202 specifically targets class I HDACs–HDAC1, HDAC2, and HDAC3– and the histone demethylase LSD1 (4SC company data: B.P.S. Bioscience Assay Report, Reaction Biology Corporation Assay Results, detailed results available upon request) [5]. Prior studies have demonstrated that 4SC-202 has strong anti-tumor activities in a number of cancer cell lines and preclinical models. For example, 4SC-202 has been demonstrated to reduce proliferation of all epithelial and mesenchymal urothelial carcinoma (UC) cell lines [6,7]. In addition, 4SC-202 has been shown to reduce proliferation and survival of human colorectal cells and inhibit growth of colorectal tumors in vivo [8]. 4SC-202 has been evaluated in a Phase I clinical trial for blood cancer, and anti-tumor efficacy has thus far been observed [9]. Based on these past studies and ongoing trials, the goal of this project was to examine the efficacy of 4SC-202 in pediatric medulloblastoma.

## 2. Materials and Methods

### 2.1. Cell Culture, Cell Viability Assays, and Immunocytochemistry

Frozen adherent desmoplastic cerebellar medulloblastoma brain cells obtained from a four-year-old Caucasian male (DAOY; ATCC-HTB-186) were cultured in DMEM with 10% FBS, Penicillin at 37 °C. To test the effect of 4SC-202 on non-cancerous cells, the following control cell line was used: neural stem cells (MTI-GlobalStem) grown in Gibco Astroycte media containing DMEM, with N-2 supplement, and One Shot FBS. Cell viability assays were conducted with Cell Titer Glo 2.0 (Promega, Madison, WI, USA) using the Glowmax 96 microplate luminometer (Promega). One thousand cells were plated per 96 well in a Corning 96 Well Solid White Flat Bottom Polystyrene TC-Treated Microplates (Cat #3917). Concentrations of 4SC-202 ranging from 0.001 µM–10 µM were applied to DAOY cells for 72 h. Control treatments consisted of DMSO at 0.001%. Cell Viability was measured using Cell Titer Glo 2.0 (Promega, Madison, WI, USA), and luminescence was measured using the Glowmax software on a Glowmax 96 microplate luminometer (Promega, Madison, WI, USA).

For immunocytochemistry, DAOY medulloblastoma American Type Culture Collection (ATCC; Manassas, VA) cells were plated on PureCol purified collagen I (Biocare Medical, Pacheco, CA, USA) imaging dishes as described [10]. One week after plating, cells were treated with 4SC-202 at 58 nM for 72 h, with control-treated cells treated with vehicle DMSO at 0.00001%. Following 72 h exposure to 4SC-202, cells were fixed with 4% paraformaldehyde for 10 min, washed 3× with 1× PBS, permeabilized with the 0.1% Triton-X for 10 min, washed 3× with 1× PBS, blocked, and stained with primary antibody against Sonic Hedgehog conjugated to Alex Fluor 488 (EP1190Y, ab203961, Abcam, Cambridge, MA, USA) diluted 1:200, and secondary antibody Goat anti-rabbit IgG with Alexa Flour 488. Cells were imaged on a confocal Olympus FV1200 Laser Scanning Microscope.

### 2.2. Spheroid Culture

One thousand DAOY cells were plated in Corning^®^ spheroid microplates and allowed to form spheroids for one week prior to being treated with 58 nm 4SC-202. After 72 h of treatment the spheroids were stained with caspase-3/7 (Cell Event Caspase-3/7 Green detection, Thermofisher, Waltham, MA, USA) and the NucRedDead 647 Ready Probe Reagent (Thermofisher, Waltham, MA, USA) and imaged using the OlympusIX71 microscope.

### 2.3. Microarray Analysis

Normalized mRNA expression values from Affymetrix HG-U133plus2 chips were extracted from the Gene Expression Omnibus (GEO) repository, dataset numbers GSE66354 and GSE35493 [11,12]. For GSE66354, gene expression data from eight SHH MBs, two normal cerebellums, three normal frontal lobes, one normal medulla, one normal midbrain, two normal occipital lobes, two normal parietal lobes, one normal temporal lobe, and one thalamus (*n* = 13 normal brain samples) were used in the analysis. For GSE35493, gene expression data from 17 MBs, 2 normal cerebellums, 1 normal frontal lobe, 2 normal temporal lobes, 2 normal occipital lobes, and 2 normal parietal lobes (n = 9 normal brain samples) were used. Each gene of interest in these datasets was identified by a single Affymetrix probe set ID except *EZH*2 and *HDAC*2, which are each identified by two probe set IDs. *EZH2* gene expression was determined based on the results from probe 203358_s_at and *HDAC2* from probe 201833_at. Error is represented as the standard error of the mean (SEM). Significance was determined using Linear Models for Microarray Data (LIMMA) package in R Bioconductor through GEO2R between MB and normal cerebellum as well as between MB and all normal brain samples [13]. *P* values were adjusted using the Benjamini and Hochberg procedure [14].

## 3. Results

### 3.1. 4SC-202 Is Cytotoxic to Medulloblastoma in Cell Culture

Exposure of the medulloblastoma cell line to 4SC-202 for 72 h significantly reduces viability at concentrations ranging from 0.001–10 µM (Figure 1), but does not significantly affect the growth of the control NSCs. Concentrations ranging from 0.001–0.3 µM were significantly different from control-treated cells, with a paired two-tailed *t* test, *p* < 0.05. Concentrations ranging from 1–10 µM were significantly different from control-treated cells, with a paired two-tailed *t* test, *p* < 0.001.

To examine the mechanism of cell death, DAOY cells and spheroids were stained with caspase-3/7, Dead Red, and Hoechst 33342 for 72 hrs. following 4SC-202 exposure. There is an increase in Dead Red staining in the 4SC-202-treated cells relative to the control sample. Additionally, the caspase-3/7 stain is colocalized with Dead Red in drug-treated cells (Figure 2).

### 3.2. 4SC-202 Inhibits Hedgehog Immunoreactivity

Exposure of the medulloblastoma cell line DAOY to 58 nM 4SC-202 blocks immunoreactivity for hedgehog. Positive staining for hedgehog was observed in control-treated DAOY cells (Figure 3B), which was reduced in 4SC-202 treated cells (Figure 3E). Reduced cell number and nuclei observed was also observed in 4SC-202 cells (Figure 3D) as compared to control-treated cells (Figure 3A).

### 3.3. Microarray Analysis of Human Medulloblastoma Tumors

Microarray analysis from human medulloblastoma patients indicates significant upregulation of the gene expression levels in medulloblastoma for a number of known target proteins for 4SC-202 and SHH pathway proteins (Figure 4). LSD1, HDAC2, and HDAC3 are upregulated in both SHH-MBs and the nonspecific MB microarray results, but there is little evidence to suggest that *HDAC1* is upregulated. SHH mediators *GLI*2 and *GLI*3 are significantly upregulated in SHH-MB, but *GLI3* is not upregulated in the larger pool of medulloblastomas, and there is only weak evidence that *GLI2* is upregulated. It is interesting to note that while *GLI*2 and *GLI*3 are upregulated, there is no significant difference in the detected number of *SHH* transcripts. Additionally, *PTK*7 and *EZH*2 are significantly upregulated for both MB groups.

## 4. Discussion

Currently, there are few effective therapies for pediatric medulloblastoma, with surgery often not able to remove residual metastastic tissue. In recent years, transcriptome profiling of human cancers has revealed important insights into the biology of tumors. In this brief report, following an analysis of microarrays of medulloblastoma patients, we identify key proteins—such as *LSD1, EZH2, GLI2, GLI3*, and *PTK7*—that may serve as drug targets. In addition, we examine the efficacy of 4SC-202 in a pediatric medulloblastoma cell line. Our studies indicate that exposure of the medulloblastoma cells to 4SC-202 significantly reduces cell viability and provides the groundwork for a future more in-depth analysis of the effect of 4SC-202 on medulloblastoma, both in vitro and in preclinical models.

Upregulation of target proteins identified by analysis of the microarray data may be important targets for future therapies for medulloblastoma. As we have previously demonstrated, upregulation of *PTK*7 may indicate that it is an important target for pediatric cancers [3]. LSD1, HDAC2, and HDAC3 are key targets for 4SC-202, and thus upregulation of these targets across the studied medulloblastomas makes 4SC-202 a more actionable therapeutic.

*GLI2* and *GLI3* are two of the principle mediators of the sonic hedgehog signaling pathway (Shh) [15] and are upregulated in SHH-medulloblastomas (SHH-MB) relative to normal brain samples. It was recently demonstrated that selectively inhibiting HDAC1 and HDAC2 chemically decreases the tumor growth of SHH-MB in a murine model by inhibiting the Hedgehog pathway (Hh), which is further linked to an increase in GLI1 acetylation [16]. Likewise, the tumor killing activity of 4SC-202 may be related to the inhibition of the Hh pathway. Alternately, in pancreatic ductal adenocarcinoma (PDAC), it was found that 4SC-202 decreases TGFβ signaling [17]. TGFβ signaling and Hh signaling converge around *GLI2* [18], and differences in the level of TGFβ signaling have been associated with slowed MB progression and improved patient outcomes, though the mechanism remains unclear [19,20,21]. Further preclinical studies and pathway studies need to be conducted to validate the efficacy and mechanism of action of 4SC-202 for in vivo models.

## Figures and Tables

**Figure 1 brainsci-07-00147-f001:**
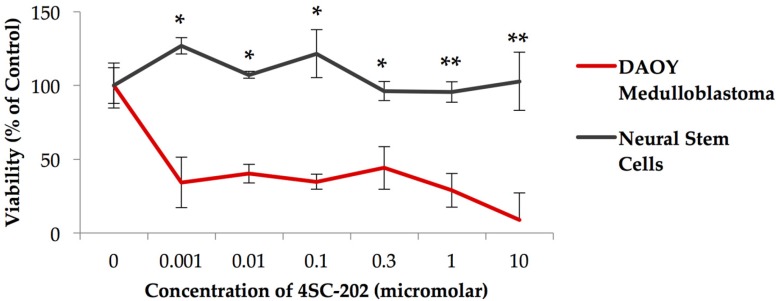
Exposure of DAOY medulloblastoma to 4SC-202 significantly reduces cell viability but does not affect the viability of control neural stem cells, with *p* value < 0.01 at concentrations ranging from 1–10 µM (**), and *p* value < 0.05 (*) at concentrations ranging from 0.001–1 µM. Viability was measured using Cell Titer Glo 2.0 (Promega).

**Figure 2 brainsci-07-00147-f002:**
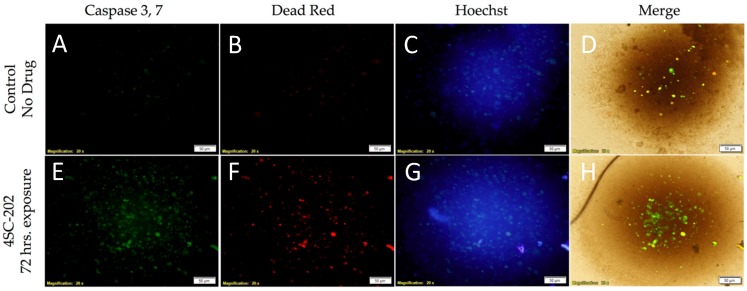
4SC-202 induces caspase-3/7 activities in DAOY spheroids. DAOY spheroids were treated with 4SC-202 for 72 h. Prior to being stained with caspase-3/7, Dead Red, and Hoechst stains. Caspase-3/7 activities are present in spheroids treated with 4SC-202 (**E**) but absent in DMSO control-treated spheroids (**A**). Increased dead cells were observed in the 4SC-202-treated spheroid (**F**) compared to DMSO control-treated spheroids (**B**). Hoechst stain visualizes nuclei in both conditions (**C**,**G**). Increased colocalization of caspase-3/7 with dead cells is visualized in (**H**) as compared to (**D**). Scale bar = 50 microns.

**Figure 3 brainsci-07-00147-f003:**
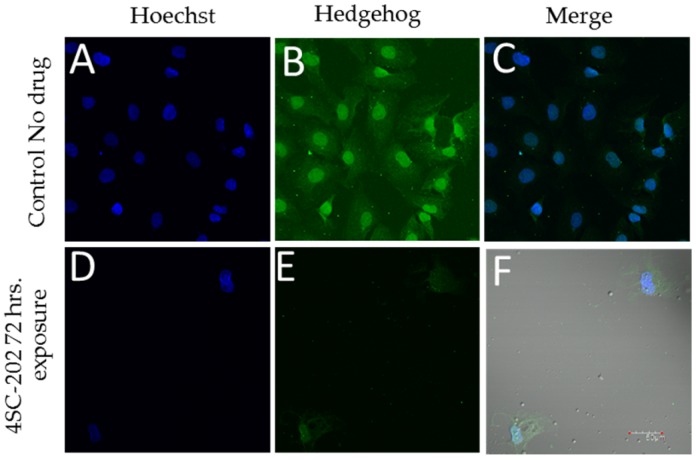
4SC-202 reduces Hedgehog expression in DAOY medulloblastoma. Hedgehog immunoreactivity is present in control (0.001% DMSO) (**B**) treated DAOY but significantly reduced in DAOY treated with 4SC-202 for 72 h. (**E**, white arrows). The number of nuclei was significantly reduced following 4SC-202 exposure due to the cytotoxicity of 4SC-202 as assessed by staining with Hoechst dye (**D**) when compared to the number of nuclei in control treated DAOY (**A**). Colocalization of Hoechst and Hedgehog immunoreactivity is illustrated in control treated DAOY (**C**) and 4SC-202 treated DAOY (**F**), showing fewer hedgehog immunoreactivity treatment following 4SC-202 treatment. Scale bar = 50 microns.

**Figure 4 brainsci-07-00147-f004:**
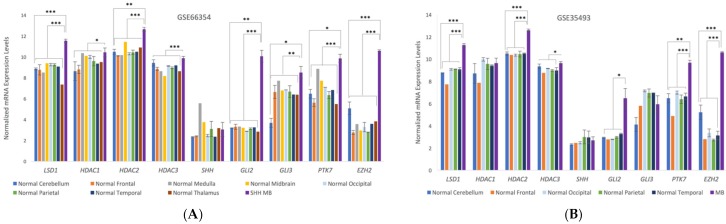
Normalized expression levels of key HDACs, SHH pathway genes, and other genes of interest in (**A**) human normal brain samples and SHH-MBs (GSE66354) based on microarray data from Griesinger et al. [11] or in (**B**) human normal brain samples and MBs of unspecified subtypes (GSE35493) based on microarray data from Birks et al. [12]. Error bars represent the standard error of the mean. Significance was tested across MBs (*n* = 8 GSE66354, *n* = 17 GSE35493) versus normal cerebellum (*n* = 2 GSE66354, *n* = 2 GSE35493) or all normal brain tissue (*n* = 13 GSE66354, *n* = 9 GSE35493). *p* values were adjusted using the Benjamini and Hochberg procedure [14]; *p* < 0.001 (***), *p* < 0.01 (**), *p* < 0.1 (*).

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
