# Peer review of "4SC-202 as a Potential Treatment for the Pediatric Brain Tumor Medulloblastoma"

_brainsci, 2017, doi:10.3390/brainsci7110147_

Round 1

Reviewer 1 Report

In the Communication titled “4SC-202 as a potential treatment for the pediatric brain tumor medulloblastoma” Messerli and Coll. describe the anti-tumoral effect of the dual modulator of both HDAC1, 2, 3 and LSD1 4SC-202, specifically in medulloblastoma cell line DAOY. Based on microarray data from Griesinger et al, the Authors also postulate some targets of the drug: most specifically, these targets were found overexpressed in SHH medulloblastoma versus control brain.

The study presents several issues:

1)     The study appear to be restricted to the SHH Medulloblastoma subgroup, so this should be specified in the title as well along all the text, avoiding to generally speaking about pediatric brain tumor medulloblastoma.

2)     LSD1 HDAC1 SHH don’t appear to be significantly differentially expressed between tumour and control samples, by performing the analysis through GEO2R (dataset number GSE66354). Authors should explain this point (see point 4 of my revision).

3)     The number of samples analyzed is scanty (only 8 SHH Medulloblastoma samples against 2 healthy control): the Authors are strongly suggested to investigate other (wider) datasets, in order to render their statements stronger. Furthermore is also suggested to verify how the targets identified by the Authors vary their expression in other pediatric tumors or in the SHH Medulloblastoma versus other healthy regions of the brain.

4)     The real rationale of the project should be better described: 4sc-202 is a dual modulator of both HDAC1, 2, 3 and LSD1. Authors should describe (if known) how the drug really works at the molecular level: this is critical to draw conclusions about the results obtained, further to contribute in understanding the potential side effects of this drug. Authors should also describe the limit of the study: HDAC modulators normally act broad-spectrum, so their effects on several physiological cell lines should be investigated.

5)      Figure 2 is not highly representative, due to the high rate of cell death revealed after 72 hours of treatment: Authors should show a time course experiment (from 12 to 72h), during which they assay Hedgehog expression. The same time course experiment should be performed in order to define the dose-response treatment

6)      In this study the effects of 4sc-202 have been observed only on proliferation. Authors are strongly suggested to assay also effects on apoptosis, differentiation and migration/invasion.

7)      Henning SW, Doblhofer R, Kohlhof H, Jankowsky R, Maier T, Beckers T, et al. Preclinical characterization of 4SC-202, a novel isotype specific HDAC inhibitor. EJC Suppl. 2010;8:61. doi: 10.1016/S1359-6349(10)71883-8 should be cited

Minor points:

1)      In the abstract (lane 17) Authors say that 4sc-202’s IC50 has been determined based on a comparison with the control astrocytes: based on Figure 1’s legend control neural stem cells (not astrocytes) have been used as control. Authors should explain this discrepancy.

2)      In the Materials and Methods section the description of dose-response experiments should be added.

3)      Even if deducible from the context, Authors should define the acronym SHH MB, firstly described at line 66.

4)      The phrase “Concentrations ranging from 0.001-0.3 μM were significantly different from control treated cells, with a paired two-tailed t test, p<0.05. Concentrations ranging from 1-10 μM were significantly different from control treated cells, with a paired two-tailed t test, p<0.001” on lanes 75 and 76 should be better explained by the Authors. Why they describe control cells as “control treated”?

5)      In lanes 91-93 Authors refer to several panels of figure 2, but these panels are not shown in the figure.

6)      All the section from lane 97 to 108 should be moved to Discussion section.

7)      At lane 97 Authors speak about upregulation of LSD1, EZH2, GLI2, GLI3, and PTK7 proteins, but they limit to describe the altered expression of mRNAs.

Author Response

Reviewer #2:

( ) I would like to sign my review report

English language and style

( ) Extensive editing of English language and style required
(x) Moderate English changes required
( ) English language and style are fine/minor spell check required
( ) I don't feel qualified to judge about the English language and style

Yes

Can be improved

Must be improved

Not applicable

Does the introduction   provide sufficient background and include all relevant references?

( )

(x)

( )

( )

Is the research design   appropriate?

( )

( )

(x)

( )

Are the methods adequately   described?

( )

( )

(x)

( )

Are the results clearly   presented?

( )

( )

(x)

( )

Are the conclusions   supported by the results?

( )

(x)

( )

( )

Comments and Suggestions for Authors

In the Communication titled “4SC-202 as a potential treatment for the pediatric brain tumor medulloblastoma” Messerli and Coll. describe the anti-tumoral effect of the dual modulator of both HDAC1, 2, 3 and LSD1 4SC-202, specifically in medulloblastoma cell line DAOY. Based on microarray data from Griesinger et al, the Authors also postulate some targets of the drug: most specifically, these targets were found overexpressed in SHH medulloblastoma versus control brain.

 The study presents several issues:

The study appear to be restricted to the SHH Medulloblastoma subgroup, so this should be specified in the title as well along all the text, avoiding to generally speaking about pediatric brain tumor medulloblastoma.

Author comment: A sentence regarding that the study appears to address SHH Medulloblastoma has been added to the discussion. However, the results may be applicable to all the Medulloblastoma subgroups, particularly given that LSD1, HDAC2, and HDAC3 are all upregulated in the larger medulloblastoma pool. This has not been fully investigated in this study, but is important to investigate in further more in an in-depth research articles.

The  LSD1 HDAC1 SHH don’t appear to be significantly differentially expressed between tumour and control samples, by performing the analysis through GEO2R (dataset number GSE66354). Authors should explain this point (see point 4 of my revision).  

Author response: LSD1 is significantly differentially expressed, but to address the issue, we also added data showing that HDAC2 and HDAC3 are also differentially expressed across two different datasets.

The number of samples analyzed is scanty (only 8 SHH Medulloblastoma samples against 2 healthy control): the Authors are strongly suggested to investigate other (wider) datasets, in order to render their statements stronger. Furthermore is also suggested to verify how the targets identified by the Authors vary their expression in other pediatric tumors or in the SHH Medulloblastoma versus other healthy regions of the brain.

These questions are valid and may be the topic of a future more in depth research article. For this Short Communication, the authors expanded the microarray analysis to include another dataset that includes multiple types of MB and specified what the gene expression values were for each region of the brain rather than just cerebellum.

4)     The real rationale of the project should be better described: 4sc-202 is a dual modulator of both HDAC1, 2, 3 and LSD1. Authors should describe (if known) how the drug really works at the molecular level: this is critical to draw conclusions about the results obtained, further to contribute in understanding the potential side effects of this drug. Authors should also describe the limit of the study: HDAC modulators normally act broad-spectrum, so their effects on several physiological cell lines should be investigated.

Since HDAC modulators are broad spectrum, he authors have investigated the effect of 4SC-202 on several physiological cell lines such as astrocytes and Neural Stem Cells (NSC) and found that 4SC-202 did not reduce cell viability.

Author response : These important point are now addressed in the Introduction and Discussion.

5)      Figure 2 is not highly representative, due to the high rate of cell death revealed after 72 hours of treatment: Authors should show a time course experiment (from 12 to 72h), during which they assay Hedgehog expression. The same time course experiment should be performed in order to define the dose-response treatment.

.Positive staining for hedgehog was observed in control treated DAOY cells which was reduced in 4SC-202 treated cells. Reduced cell number and nuclei observed in 4SC-202 cells as compared to control treated cells. More detailed time course experiment examining how hedgehog expression changes will be presented in a detailed research paper, but is outside the scope of this Short Communication.

6)      In this study the effects of 4sc-202 have been observed only on proliferation. Authors are strongly suggested to assay also effects on apoptosis, differentiation and migration/invasion. Unpublished data suggests 4SC-202 may induce apoptosis in Medulloblastoma. However, since this is a Short Communication, authors prefer not to describe apoptosis, differentiation, and migration/invasion as these studies are ongoing and the subject of a full research article.

7)      Henning SW, Doblhofer R, Kohlhof H, Jankowsky R, Maier T, Beckers T, et al. Preclinical characterization of 4SC-202, a novel isotype specific HDAC inhibitor. EJC Suppl. 2010;8:61. doi: 10.1016/S1359-6349(10)71883-8 should be cited.

This reference, which was the first publication about 4SC-202 has been cited in the introduction..

Minor points:

1)      In the abstract (lane 17) Authors say that 4sc-202’s IC50 has been determined based on a comparison with the control astrocytes: based on Figure 1’s legend control neural stem cells (not astrocytes) have been used as control. Authors should explain this discrepancy.

Although both astrocytes and neural stem cells were used as controls, the graph depicts the viability of NSC’s in response to 4SC-202 treatment, This typo in the Abstract has been corrected.

2)      In the Materials and Methods section the description of dose-response experiments should be added.

This description has been added.

3)      Even if deducible from the context, Authors should define the acronym SHH MB, firstly described at line 66.  This definition has been added.

4)      The phrase “Concentrations ranging from 0.001-0.3 μM were significantly different from control treated cells, with a paired two-tailed t test, p<0.05. Concentrations ranging from 1-10 μM were significantly different from control treated cells, with a paired two-tailed t test, p<0.001” on lanes 75 and 76 should be better explained by the Authors. Why they describe control cells as “control treated”?

The controls consisted of equal concentrations of DMSO as in the drug treatments. This has now been explained in the methods and made more clear in the sentence above.

5)      In lanes 91-93 Authors refer to several panels of figure 2, but these panels are not shown in the figure.

This has been corrected.

6)      All the section from lane 97 to 108 should be moved to Discussion section. These sentences from the Results section have been moved to the Discussion

7)      At lane 97 Authors speak about upregulation of LSD1, EZH2, GLI2, GLI3, and PTK7 proteins, but they limit to describe the altered expression of mRNAs.

This sentence has been changed to :upregulation of LSD1, EZH2, GLI1, GLI3, and PTK7 RNA.

Reviewer 2 Report

In this manuscript (BrainSci 225929), the authors have briefly presented the effect of a small molecule 4SC-202 on the cell viability of medulloblastoma cell lines and on the hedgehog expression. Although the cell viability and the hedgehog inhibition results are somewhat interesting, the manuscript has several problems,

1)     4SC-202 is a drug candidate being developed by the company 4SC, Germany. One of the authors, Hella Kohlhof seems to be part of the company 4SC. However, the authors have claimed “no conflict of interest” and they have not mentioned anywhere in the manuscript that this is a study of their own compound.

2)     Line 30 - the authors claim that 4SC-202 is a “novel orally available benzamide type HDAC inhibitor (HDACi) specific for class I HDACs, HDAC1, HDAC2 and HDAC3, and the histone demethylase LSD1”. However, no reference has been given to support their claim.

3)     Line 35 - “4SC-202 has been demonstrated to reduce proliferation of all epithelial and mesenchymal urothelial carcinoma (UC) cell lines [5]”. However, the name 4SC-202 could not be located anywhere in ref. 5. Is 4SC-202 called by other name(s)?

4)     Line 38 - “4SC-202 has been evaluated in Phase I clinical trials for blood cancer, and anti-tumor efficacy has thus far been observed (http://www.4sc.com/product-pipeline/4sc-202)”. A company webpage cannot be considered as a scientific article.

5)     Line 99 – “LSD1 is a key target for 4SC-202”. Again a literature reference is needed here to support this claim.

6)     Line 129 – Author contributions – It says M.H.H analyzed the data, But there is no author listed with initials M.H.H.

7)     Line 130 – Similarly, it says EZ contributed to bioinformatics analysis. But there is no author with initials E.Z. Also it says M.H. wrote paper. Are these typos?

8)     Author contribution – The author, Hella Kohlhof’s contributions are not listed.

Author Response

Reviewer # 1:

Reviewer Comment: 4SC-202 is a drug candidate being developed by the company 4SC, Germany. One of the authors, Hella Kohlhof seems to be part of the company 4SC. However, the authors have claimed “no conflict of interest” and they have not mentioned anywhere in the manuscript that this is a study of their own compound.

Author Response: We have added the following to the ‘Conflicts of Interest’ section: H.K. was an employee of 4SC AG until 31 December, 2016. H.K. is currently affiliated with Immunic AG. 4SC-202 is still the property of 4SC-202 and clinical trials are ongoing. H.K. holds a minor amount of shares in 4SC AG. Other authors claim no conflict of interest.

2)    Reviewer Comment Line 30 - the authors claim that 4SC-202 is a “novel orally available benzamide type HDAC inhibitor (HDACi) specific for class I HDACs, HDAC1, HDAC2 and HDAC3, and the histone demethylase LSD1”. However, no reference has been given to support their claim.

Author Comment: The Reference Pinkerneil et al. 2016 has been inserted.

Reviewer Comment: Line 35 - “4SC-202 has been demonstrated to reduce proliferation of all epithelial and mesenchymal urothelial carcinoma (UC) cell lines [5]”. However, the name 4SC-202 could not be located anywhere in ref. 5. Is 4SC-202 called by other name(s)?

Author Comment: The publication Pinkerneil et al. 2016 is referenced  and 4SC-202 is mentioned quite often. Might have been a misunderstanding with the reviewers.

Reviewer Comment: Line 38 - “4SC-202 has been evaluated in Phase I clinical trials for blood cancer, and anti-tumor efficacy has thus far been observed (http://www.4sc.com/product-pipeline/4sc-202)”. A company webpage cannot be considered as a scientific article.

Author Comments: The reference: Tresckow B, Gundermann S, Eichenauer DA, Aulitzky WE, Göbeler M, Sayehli C et al. First-in-human study of 4SC-202, a novel oral HDAC inhibitor in advanced hematologic malignancies (TOPAS study). Journal of Clinical Oncology. 2014;32(supplement 5):abstr. 8559 has been added.

Reviewer Comment:

Line 99 – “LSD1 is a key target for 4SC-202”. Again a literature reference is needed here to support this claim.

Author Response: The following reference has been added to support this claim.

Kohlhof, H., Gruber, W., Vitt, D., Aberger, F. & Prenzel, T. Abstract C89: The small molecule inhibitor 4SC-202 controls aberrant HH signaling in cancer. Mol Cancer Ther 14, C89–C89 (2015).

Reviewer Comment:     Line 129 – Author contributions – It says M.H.H analyzed the data, But there is no author listed with initials M.H.H.

Author comment: The correct initials M.H.H. has been added for this author.

Reviewer Comment: Line 130 – Similarly, it says EZ contributed to bioinformatics analysis. But there is no author with initials E.Z. Also it says M.H. wrote paper. Are these typos?

Author Comment: These typos have been corrected.

Reviewer Comment: Author contribution – The author, Hella Kohlhof’s contributions are not listed.

Author Comment: HK contributed to experiment design, data discussion and interpretation, paper preparation

Round 2

Reviewer 1 Report

The Authors improved the paper. A better resolution of figure 2 is needed. In Figure 4's legend the number of samples (n) considered for statistical analysis should be added.

Author Response

The Authors improved the paper. A better resolution of figure 2 is needed. In Figure 4's legend the number of samples (n) considered for statistical analysis should be added.

The resolution of Figure 2 was improved.

The number of samples used for the statistical analyses were included in the figure legend for Figure 4 (lines 123-124). “Significance was tested across MBs (n=8 GSE66354, n=17 GSE35493) versus normal cerebellum (n=2 GSE66354, n=2 GSE35493) or all normal brain tissue (n=13 GSE66354, n=9 GSE35493).”                                                    

Reviewer 2 Report

In the revised manuscript, BrainSci-225929-v2, the authors have satisfactorily answered only to some of the questions raised by reviewer 1. The following issues still remain unanswered.

1)      Reviewer1 comment to original manuscript line 30 - “…………… the authors claim that 4SC-202 is a “novel orally available benzamide type HDAC inhibitor (HDACi) ……….”. However, no reference has been given to support their claim.

Author Comment: The Reference Pinkerneil et al. 2016 has been inserted”
Revised manuscript line 58

The authors say the reference Pinkerneil et al. 2016 has been inserted. However, the authors have actually replaced the existing Pinkerneil et al., Mol. Cancer Ther. 2016, 15, 299–312 reference with Pinkerneil et al., Target. Oncol. 2016, 11, 783–798. Even so, the modified Pinkerneil et al. 2016 reference provided with revised manuscript only has the following information regarding the HDAC inhibition activity of 4SC-202.  

Target. Oncol. 2016, 11, 783-798, Page 784, Column 1 - “4SC-202 is a benzamide type inhibitor with strong activity against HDAC1 (IC50: 0.16 μM), HDAC2 (0.37 μM) and HDAC3 (0.13 μM), without affecting other HDAC enzymes at clinically relevant concentrations (IC50: HDAC4, HDAC5, HDAC6, HDAC7, HDAC8, HDAC9, HDAC10, HDAC11 > 15 μM) (updated, unpublished data, personal communication by H.K., detailed data available upon request).  The reported IC50 for KDM1A/LSD1 ranges in clinically relevant concentrations from 0.6 to 1.2 μM (Data presented at Sixth Annual EpiCongress, Boston, USA, July 2015, data available online at https://4SC.de).”

Other than the above information the article does not provide any experimental data/details for HDAC inhibition of 4SC-202.  Reference to a scientific article giving the experimental data with details is required to show HDAC inhibitory property of 4SC-202. Without supporting experimental details, the HDAC inhibition by 4SC-202 can only be stated as a hypothesis and not as a fact.

2)      Reviewer1 comment to original manuscript line 99 – “LSD1 is a key target for 4SC-202”. Again a literature reference is needed here to support this claim.
Author Response: The following reference has been added to support this claim.

Kohlhof, H., Gruber, W., Vitt, D., Aberger, F. & Prenzel, T. Abstract C89: The small molecule inhibitor 4SC-202 controls aberrant HH signaling in cancer. Mol Cancer Ther 14, C89–C89 (2015).”
Revised manuscript line 191 – “LSD1, HDAC2 and HDAC3 are key targets for 4SC-202”

The authors say that they have included the above reference in the revised manuscript. However, it appears there was no reference added in the revised manuscript. Nevertheless, the above mentioned abstract does not provide any LSD1 inhibition experimental data for 4SC-202. Without supporting experimental data, the LSD1, HDAC2 and HDAC3 as key targets for 4SC-202 can only be stated as a hypothesis.

3)      Reviewer1 comment to original manuscript line 35 - “4SC-202 has been demonstrated to reduce proliferation of all epithelial and mesenchymal urothelial carcinoma (UC) cell lines [5]”. However, the name 4SC-202 could not be located anywhere in ref. 5. Is 4SC-202 called by other name(s)?
Author Comment: The publication Pinkerneil et al. 2016 is referenced and 4SC-202 is mentioned quite often. Might have been a misunderstanding with the reviewers.
Revised manuscript line 63

The original manuscript ref. 5 was Pinkerneil et al., Mol. Cancer Ther. 2016, 15, 299–312. This reference did not have anything about 4SC-202. In the revised manuscript ref. 5, the authors removed this reference and included Pinkerneil et al., Target. Oncol. 2016, 11, 783–798. So apparently in the original manuscript, the authors have made a mistake by including the wrong article by Pinkerneil et al. The reviewer did not have a misunderstanding about the article listed as ref. 5 in the original manuscript.

Author Response

In the revised manuscript, BrainSci-225929-v2, the authors have satisfactorily answered only to some of the questions raised by reviewer 1. The following issues still remain unanswered. 1)      Reviewer1 comment to original manuscript line 30 - “…………… the authors claim that 4SC-202 is a “novel orally available benzamide type HDAC inhibitor (HDACi) ……….”. However, no reference has been given to support their claim. Author Comment: The Reference Pinkerneil et al. 2016 has been inserted”
The authors say the reference Pinkerneil et al. 2016 has been inserted. However, the authors have actually replaced the existing Pinkerneil et al., Mol. Cancer Ther. 2016, 15, 299–312 reference with Pinkerneil et al., Target. Oncol. 2016, 11, 783–798. Even so, the modified Pinkerneil et al. 2016 reference provided with revised manuscript only has the following information regarding the HDAC inhibition activity of 4SC-202.  

Target. Oncol. 2016, 11, 783-798, Page 784, Column 1 - “4SC-202 is a benzamide type inhibitor with strong activity against HDAC1 (IC50: 0.16 μM), HDAC2 (0.37 μM) and HDAC3 (0.13 μM), without affecting other HDAC enzymes at clinically relevant concentrations (IC50: HDAC4, HDAC5, HDAC6, HDAC7, HDAC8, HDAC9, HDAC10, HDAC11 > 15 μM) (updated, unpublished data, personal communication by H.K., detailed data available upon request).  The reported IC50 for KDM1A/LSD1 ranges in clinically relevant concentrations from 0.6 to 1.2 μM (Data presented at Sixth Annual EpiCongress, Boston, USA, July 2015, data available online at https://4SC.de).”

Other than the above information the article does not provide any experimental data/details for HDAC inhibition of 4SC-202.  Reference to a scientific article giving the experimental data with details is required to show HDAC inhibitory property of 4SC-202. Without supporting experimental details, the HDAC inhibition by 4SC-202 can only be stated as a hypothesis and not as a fact.

4SC did not publish all data regarding 4SC-202 in peer reviewed journals. Some data were only published on posters or cited as personal communications. We attached the original report from the service provider (Reaction Biology) performing the HDAC activity assay, including the methods. The code for 4SC-202 in this report is SC85956. There are further compounds included which are not relevant for this publication. The experimental data to support LSD1 inhibition are included in the report from the service provider BPS Bioscience. 4SC-202 is coded as SC 00085956. (Please see as well response for 2). 

2)      Reviewer1 comment to original manuscript line 99 –LSD1 is a key target for 4SC-202”. Again a literature reference is needed here to support this claim.
Author Response: The following reference has been added to support this claim.

Kohlhof, H., Gruber, W., Vitt, D., Aberger, F. & Prenzel, T. Abstract C89: The small molecule inhibitor 4SC-202 controls aberrant HH signaling in cancer. Mol Cancer Ther 14, C89–C89 (2015).”Revised manuscript line 191 – “LSD1, HDAC2 and HDAC3 are key targets for 4SC-202”

The authors say that they have included the above reference in the revised manuscript. However, it appears there was no reference added in the revised manuscript. Nevertheless, the above mentioned abstract does not provide any LSD1 inhibition experimental data for 4SC-202. Without supporting experimental data, the LSD1, HDAC2 and HDAC3 as key targets for 4SC-202 can only be stated as a hypothesis.

We attached the pdf of the poster that was presented according to the reference given above. There, the data for HDAC and LSD1 inhibition are demonstrated.  These data were not published in a peer reviewed journal, but only on this poster. We attached the original report from the service provider (BPS Bioscience) performing the LSD1 activity assay, including the methods. The code for 4SC-202 in this report is SC 00085956. The report supporting the HDAC inhibition is the same as mentioned in comment 1 from reviewer 1. 

The poster reference was missing and is now reference [5].  To clarify the difference between peer-reviewed and company data, in the revised manuscript, the poster and assays are mentioned as follows:

(4SC company data: B.P.S. Bioscience Assay Report, Reaction Biology Corporation Assay Results, Kohlhof et al., 2015 [5], detailed results available upon request)” (lines 38-39).

3)      Reviewer1 comment to original manuscript line 35 - “4SC-202 has been demonstrated to reduce proliferation of all epithelial and mesenchymal urothelial carcinoma (UC) cell lines [5]”. However, the name 4SC-202 could not be located anywhere in ref. 5. Is 4SC-202 called by other name(s)?
Author Comment: The publication Pinkerneil et al. 2016 is referenced and 4SC-202 is mentioned quite often. Might have been a misunderstanding with the reviewers.
Revised manuscript line 63

The original manuscript ref. 5 was Pinkerneil et al., Mol. Cancer Ther. 2016, 15, 299–312. This reference did not have anything about 4SC-202. In the revised manuscript ref. 5, the authors removed this reference and included Pinkerneil et al., Target. Oncol. 2016, 11, 783–798. So apparently in the original manuscript, the authors have made a mistake by including the wrong article by Pinkerneil et al. The reviewer did not have a misunderstanding about the article listed as ref. 5 in the original manuscript.

as ref. 5 in the original manuscript.

We apologize and you are absolutely correct. This was a mistake from our side, because we included the wrong publication. Sorry again.
